# Recent Studies on the Effects of Static Magnetic Fields (SMF) on Reproductive Function

**DOI:** 10.3390/cimb47020116

**Published:** 2025-02-11

**Authors:** Chengchang Zhang, Chengle Dong, Xiaohang Liu, Jiaxing Zhang, Qinlan Li, Shuting Chen, Hu Zhao, Donghui Huang

**Affiliations:** 1Institute of Reproduction Health Research, Tongji Medical College, Huazhong University of Science and Technology, Wuhan 430030, China; u202110370@hust.edu.cn (C.Z.); dongchenle@163.com (C.D.); u202110325@hust.edu.cn (X.L.); u202210285@hust.edu.cn (J.Z.); u202010354@hust.edu.cn (Q.L.); u202010356@hust.edu.cn (S.C.); 2Department of Human Anatomy, Tongji Medical College, Huazhong University of Science and Technology, Wuhan 430030, China; 3National Demonstration Center for Experimental Basic Medical Education, Huazhong University of Science and Technology, Wuhan 430030, China; 4Shenzhen Huazhong University of Science and Technology Research Institute, Shenzhen 518109, China

**Keywords:** SMFs, reproduction, sperm, embryos, oocyte

## Abstract

Background: With the widespread use of static magnetic fields (SMFs) in applications such as magnetic resonance imaging (MRI) and electric vehicles, concerns have arisen regarding their potential effects on reproductive health. Despite increasing research, the impact of SMFs on reproductive function remains a subject of debate, requiring further exploration. Methods: This review synthesizes animal and clinical studies on the effects of SMF on reproductive function. It examines various SMF intensities and exposure durations, focusing on mitochondrial function, chromosomal division, and embryonic development. Results: The review reveals that low-intensity SMF exposure adversely affects mitochondrial function in sperm and eggs, reducing their activity. It also impacts follicular cells, delaying chromosomal division. Medium- and high-intensity SMF exposure shows mixed results, with both potential benefits and risks, requiring further research. High-intensity SMFs may pose teratogenic risks to embryos and delay the development of fertilized eggs. The position of SMF exposure also matters, likely due to field non-uniformity. Conclusions: This review provides a foundation for further investigation into the effects of SMFs on reproductive function, highlighting the need for more comprehensive studies to assess safety and applications. Special caution is advised for pregnant women regarding SMF exposure, given its potential risks.

## 1. Introduction

With the advancement in technology, humans are increasingly exposed to static magnetic fields (SMFs) of varying intensities in daily life, particularly in medical, food, and industrial applications [1,2,3]. In medicine, magnetic resonance imaging (MRI) has seen a significant increase in field strength over the past few decades, with most hospitals currently using MRI with field strengths between 0.5T and 3T, and MRI with higher field strengths are under development [4,5]. Research indicates that MRI can cause a slight increase in systolic blood pressure and impair cognitive and sensory functions [6,7,8]. The study by Basheer et al. showed that glucose, creatinine, uric acid, and urea levels in MRI workers increased with age, suggesting that prolonged exposure to MRI may damage kidney function [9]. Therefore, the specific effects and mechanisms of SMFs on the human body remain to be unraveled.

Currently, increasing data suggest that SMFs have negative effects on the reproductive processes of both humans and animals [10]. The study by Rockhold indicates that, while the effects of SMFs on healthy adults may not be significant, it can cause irreversible damage to embryos [11]. The study by Ray shows that exposure to MRI environments during pregnancy does not increase fetal mortality but does raise the risk of the fetus developing rheumatic diseases and skin inflammation [12]. Aside from the effects on mature embryos, any disruption in the development of male sperm, from spermatogonia (2N diploid), primary spermatocytes (2N), secondary spermatocytes (1N haploid), round spermatids (1N), and elongated spermatids (1N) to sperm (1N), involving processes like mitosis, meiosis, and cell differentiation, can lead to fertilization failure or abnormal embryo development, ultimately causing infertility [13,14]. Therefore, it can be considered that male reproductive organs are among the most sensitive to external stimuli. The research by Li and Long shows that even slight environmental changes can lead to a decline in sperm quality and motility, and exposure to magnetic fields may directly result in infertility [15,16]. Similarly, the study by Telfer found that the development of female eggs is also highly susceptible to external disturbances [17].

Therefore, exploring the effects and mechanisms of SMFs on various processes of human reproduction is crucial. Previous studies have primarily summarized the effects of medium- and high-intensity static magnetic fields (SMFs) on animals and humans [18]. In contrast, this article broadens the scope to encompass research on low-intensity SMFs, along with the latest studies published between 2022 and 2024. It provides a comprehensive review of the definition of SMFs, its effects on male and female reproductive systems, and the underlying mechanisms, offering valuable insights and guidance for addressing and managing related challenges in the future.

## 2. Definition of SMFs

An SMF refers to a magnetic field whose intensity and direction do not change over time [19]. It can be classified based on field strength into ultra-high (greater than 5 T) SMFs, high-intensity magnetic fields (greater than 1 T), medium magnetic fields (1 mT to 1 T), and low magnetic fields (less than 1 mT) [20,21]. Humans are exposed to magnetic fields throughout their lives, because Earth itself is a natural SMF [22]. The geomagnetic field’s strength is approximately 50–60 μT, and while this value is not large, it plays a critical role in maintaining Earth’s environment. Research by Kistler indicates that the geomagnetic field can deflect the powerful charged particle streams (solar wind) emitted by the sun. These charged particles are diverted by the geomagnetic field, helping to preserve Earth’s atmospheric composition [23]. Additionally, the biological effects of the geomagnetic field should not be overlooked. Previous studies have shown that animals, such as pigeons and ants, use the geomagnetic field for navigation and orientation [24,25,26]. The possible mechanism involves free radicals—magnetically sensitive chemical intermediates formed by the light excitation of cryptochrome proteins in the retina [27]. SMFs also exhibit strong biological effects. It has been demonstrated that SMFs can enhance cranial bone regeneration in rats [28,29,30]. Recent studies have also found that SMFs may affect reproductive functions (Table 1).

## 3. Effects of SMFs on Mating Behavior

As mating behavior is the starting point of the embryonic development process, exploring the effects of SMFs on this behavior is a necessary foundation for studying the subsequent reproductive toxicity of SMF. However, only a few studies have so far evaluated the impact of SMF exposure on mating behavior. Nishimura and colleagues exposed male and female rats to a 0.2 mT SMF for 22 h per day, and after mating, pregnant rats continued to be exposed to an SMF for seven days. They then compared the fertility index and mating frequency between groups and found no statistically significant differences [31]. In a study on mice continuously exposed to an SMF for about 7 days, it was found that exposure to a 4 T SMF inhibited mating behavior. However, for individuals that successfully mated, SMF exposure, even when continued until the 18th day of pregnancy, did not affect the development of their offspring [32]. Another study by Panagopoulos on *Drosophila melanogaster* exposed to an SMF showed that exposure to a 5 mT strong SMF continuously during the first 5 days of their adult lives reduced mating and reproductive behavior by 10.7%. Further research by the team revealed that SMFs are harmful to reproduction in Drosophila by inducing deoxyribonucleic acid (DNA) damage in reproductive cells [33]. Due to the size limitations of magnetic field generators, it is currently not possible to expose large groups of rats to the same magnetic field to study their mating behavior. Existing studies have small sample sizes, which may lead to errors and biases. Since *Drosophila melanogaster* have significantly different mating behaviors compared to humans, the applicability of such studies to humans is limited. Therefore, the impact of SMF exposure on human mating behavior requires further research, and no significant effects have been found thus far.

## 4. Effects of SMFs on Male Reproduction

### 4.1. Effects on Testes and Epididymis Tissues

Given that chromosomal abnormalities and genetic mutations in reproductive cells can be inherited, the reproductive toxicity of SMFs on testes and epididymis tissues is a major concern. Several studies have found significant damage to reproductive organs due to SMF exposure. It has been observed that, while exposure to a 25 μT SMF for 90 days did not significantly affect body and testicular weight in rats, it led to a decrease in the weight of seminal vesicles and preputial glands, along with a reduced sperm count [38]. Exposure to a 1.5 T SMF for 15 min has been shown to significantly decrease sperm count in mice, while inducing only slight, statistically insignificant abnormalities in sperm morphology [34]. However, after exposing mice to an MRI device for 36 min weekly over a three-week period, a reduction in testicular mass was observed, along with structural damage to the testicular parenchyma and a decrease in sperm motility [35]. Withers exposed mice to 0.3 T MRI for 66 h and found no evidence of cytotoxicity. Since this exposure duration significantly exceeds human occupational exposure, the author concluded that MRI does not negatively affect spermatogenesis or sperm development [36].

Studies indicate that male rats exposed to a 25 mT static magnetic field (SMF) for 18 consecutive weeks show a significant increase in serum luteinizing hormone (LH) levels, while testosterone levels decrease significantly [37]. After exposing *Caenorhabditis elegans* to a strong 8.5 T SMF, there was a significant increase in the apoptosis of reproductive cells. However, the apoptosis was reduced with the application of dimethyl sulfoxide (DMSO), which removes free radicals from cells. This experiment suggests that the damage to reproductive cells caused by strong SMFs may occur through oxidative stress [39]. Research suggests that SMF exposure triggers a notable rise in reactive oxygen species (ROS) levels within cells, leading to oxidative stress. This led to lipid peroxidation damage to the cell membranes, ultimately resulting in cell death [76]. A large number of studies have shown that various types of SMFs can affect intracellular ROS levels to different degrees. For example, Chao Song and colleagues found that exposure to a 0.1–0.2 T SMF can reduce liver inflammation and ROS levels [77,78,79]. Lee and colleagues suggest that SMFs damage male reproductive organs by inducing DNA damage in interstitial cells and reducing testosterone levels, leading to testicular damage [80].

Overall, high-intensity SMFs have a damaging effect on male reproductive organs, primarily by increasing ROS levels in the testes and epididymis, which induces cell apoptosis and impairs sperm production and hormone secretion. This is accompanied by an increase in gonadotropin levels. Therefore, reducing ROS production can help alleviate SMF-induced reproductive toxicity. Previous research suggests that the damage to reproductive cells caused by SMFs can be mitigated or protected by using specific antioxidants [64]. However, Donnelly’s research indicates that high concentrations of antioxidants may actually reduce sperm quality [81]. Therefore, antioxidants have potential therapeutic effects for men exposed to SMF, but the dosage must be tailored to the individual.

### 4.2. Effects on Sperm Quality

Sperm motility and morphology are critical factors that significantly influence the fertilization process [82]. Findings suggest that when exposed to a 1 T SMF in vitro, male sperm exhibit magnetotaxis, aligning their long axis perpendicular to the magnetic field [40]. Similarly, bull sperm also exhibited alignment perpendicular to the magnetic field when exposed to a 1 T SMF in vitro [42]. Exposure to a 1 mT SMF for 1 h in vitro has been linked to the weakened ability of pig sperm to undergo the acrosome reaction [43]. Yang observed a significant reduction in sperm motility in *Caenorhabditis elegans* after exposure to a 10 T SMF, along with the occurrence of sperm abnormalities in some specimens [44]. After exposing mice to a 0.7 T SMF for 35 days, 1–2 h per day, Tablado found that the hook shape of their sperm heads became abnormal, leading to structural changes [45].

The disruption of calcium ion homeostasis has been identified as a potential mechanism by which in vitro exposure to a 1 mT SMF negatively affects pig sperm motility and morphology [83]. Muti observed that, after human sperm were exposed to a 1 mT SMF for two hours in vitro, their motility significantly decreased, accompanied by a notable increase in reactive oxygen species [41]. Similarly, other research found that after exposing coral sperm to a 1 mT SMF for three hours, fertilization rates decreased, and DNA integrity was compromised [46].

The proposed mechanisms for SMF-induced damage to sperm function and DNA integrity include magnetic anisotropy of the DNA in the sperm head, the disruption of calcium ion homeostasis, and the accumulation of reactive oxygen species [40,83]. However, a few studies suggest that SMF exposure does not affect sperm motility or viability, likely due to variations in SMF intensity and exposure methods.

## 5. Effects of SMFs on Female Reproduction

### 5.1. Effects on Ovarian Function

The ovaries, responsible for producing eggs and secreting hormones, are critical for reproductive processes. When ovarian function is impaired, reproduction cannot proceed smoothly. In a previous study, adult female rats exposed to a 25 μT SMF for 18 weeks showed a significant reduction in both absolute and relative ovarian weight. Additionally, after just six weeks of exposure, the levels of gonadotropins (follicle-stimulating hormone (FSH and LH)) were significantly reduced, indicating an adverse effect on hormonal regulation [47]. In a study by Nakahara on CHO-K1 ovarian cells from female hamsters, it was found that exposure to a 10 T SMF for four days did not affect the cell growth rate or cell cycle distribution. SMF exposure alone did not increase the micronucleus frequency, but when combined with X-ray radiation, the 10 T SMF significantly increased the micronucleus frequency. It is important to note that the SMF used in this experiment was non-uniform, with the strength reaching 10 T at the center and gradually decreasing outward [48]. A 24 h exposure to a 5 mT SMF has been reported to slow the movement of entire chromosomes in hamster ovarian cells [49]. In addition to the harmful effects of SMFs on follicles, some studies suggest that SMFs may have positive effects on ovarian function. Research by Xingxing Yang and colleagues demonstrated that exposing female mice to a 150 mT SMF for 18 weeks, from 8 to 26 weeks of age, increased the number of follicles and enhanced antioxidant capacity. It also improved the abundance of the ovarian, uterine, and gut microbiomes, as well as physical performance. The researchers believe this reveals the potential of moderate SMF exposure for clinical applications [50]. In this experiment, to fully expose the mice to the SMF, eight magnets with a strength of 150 mT and the same polarity (N or S) were used to create a fixed-direction SMF. Therefore, it can be considered that the SMF used in this experiment is essentially uniform [50].

The above studies indicate that different intensities of SMF may have varying effects on ovarian function. Low-intensity SMFs can damage ovarian function and lead to decreased hormone levels in rats, while moderate-intensity SMFs can enhance the physical activity of adult female mice and improve the antioxidant capacity of the ovaries. High-intensity SMF exposure does not affect the growth and development of ovarian cells; however, when combined with X-ray exposure, it increases the frequency of micronuclei in ovarian cells, indicating increased chromosomal abnormalities.

From the perspective of experimental conditions, the intensity of the magnetic field is a crucial influencing factor, with varying effects on ovarian tissue depending on the SMF strength. Additionally, the magnetic field generators used in these experiments differ, making it difficult to rule out the impact of physical factors, such as the uniformity of the magnetic field, on the results. Similar findings in experiments on other reproductive systems also suggest that magnetic field uniformity significantly affects outcomes. Currently, no experiments have been conducted in which the intensity is significantly varied using the same magnetic field generator, while all other variables are kept constant. Moreover, due to the lack of research in this area, conclusions from individual studies may contain biases. Therefore, more experimental data are needed in the future to further determine the effects and mechanisms of SMF.

### 5.2. Effects on Oocyte Quality

Current research on the effects of SMFs on oocyte quality remains contentious. Liu exposed fertilized zebrafish eggs to a continuous 11.4 T SMF and found no significant changes in mortality, hatching rates, or body length. However, RNA sequencing analysis revealed that SMF exposure upregulated tumor necrosis factor (TNF) levels and activated the TNF signaling pathway in the zebrafish eggs [51]. It was reported that exposure to a 2 mT SMF decreased mitochondrial activity in follicular cells, subsequently impairing the reproductive function of oocytes [52]. The reorganization of cortical pigment has been noted in African clawed frog eggs following exposure to SMFs ranging from 0.5 to 9.4 T for two hours [54]. While the 11.4 T SMF did not have a significant impact on the functionality of fertilized zebrafish eggs, the implications of the elevated TNF levels remain to be further explored [51].

Existing research has demonstrated that SMFs can influence the ion concentrations and mitochondrial activity in porcine oocytes and granulosa cells through the magnetic anisotropy of lipids, leading to calcium accumulation in porcine granulosa cells and a decrease in mitochondrial activity [52,53]. Gioia et al. found that, after 72 h of exposure to a 2 mT SMF, mitochondrial activity in porcine GCs significantly weakened, resulting in a marked reduction in progesterone and estrogen production, negatively impacting reproduction and adversely affecting the proliferation, morphology, biochemical, and endocrine functions of porcine follicular granulosa cells [53]. A brief exposure of porcine follicular granulosa cells to a 2 mT SMF has been shown to induce a reversible membrane depolarization wave lasting about one minute, leading to elevated intracellular calcium levels and decreased mitochondrial activity [52].

However, another study showed that the application of a 20 mT SMF helps prevent freezing damage to oocytes, thereby aiding in their preservation [55]. Similarly, Baniasadi discovered that SMFs have a beneficial effect on the vitrification of oocytes in mature mice and can inhibit freezing damage. In exposure to a 60 mT SMF, mouse oocytes also showed reduced freezing damage and improved functionality, along with the regulation of the pluripotency of derived blastocysts [56].

The 2 mT static magnetic field (SMF) leads to ion accumulation in the mitochondria of granulosa cells and a decrease in mitochondrial activity, resulting in the diminished reproductive function of follicles. While the 11.4 T SMF currently shows no significant phenotypic effects on oocytes, further research is needed to explore its impact on oocyte quality, especially regarding the upregulation of tumor necrosis factors. However, the protective effects of SMFs ranging from 20 mT to 60 mT on cryopreserved oocytes suggest potential clinical applications for SMF. Despite these findings, it is still advisable for women planning to conceive to minimize their exposure to SMF.

## 6. Effects of SMFs on Early Embryonic Development

Multiple studies have found that SMFs can have detrimental effects on early embryonic development. Pan investigated the development time of fresh mosquito eggs, exposing them to SMFs of 9.4 T and 14.1 T. The results indicated that the 9.4 T SMF delayed the hatching time of the mosquitoes by 32 h, while the 14.1 T SMF delayed hatching by 71 h [84]. In a study by Sun et al., an in vivo analysis of medaka fish embryos under SMF exposure showed that prolonged exposure to approximately 100 mT had no significant impact on embryonic development [85]. Researchers discovered that exposing sea urchin embryos to SMFs in the range of 3.4–8.8 mT and 2.5–6.5 mT could alter early embryonic development by inducing changes in cell cycle duration, leading to delays in mitosis [86]. Exposure to an 8 T SMF has been observed to alter the orientation of the third cleavage furrow in fertilized African clawed frog embryos, shifting it from horizontal to vertical; however, this did not affect subsequent development after cleavage [60]. Ge investigated zebrafish eggs and found that, although exposing them to a 9 T SMF for 24 h over six days did not compromise the embryos’ ultimate ability to survive and develop, it did slow down the pace of their development. Notably, once the SMF was removed, the development rate of the experimental group caught up with that of the control group, indicating that the delay is at least partially reversible [61]. Research indicates that ultra-high SMF exposure at 4 T, 10 T, 14 T, and 27 T reduces the developmental speed and lifespan of *Caenorhabditis elegans* embryos [21].

In studies examining the delayed development of zebrafish exposed to SMF, it has been found that SMFs interfere with the process of mitosis during early cleavage by disrupting microtubule and spindle positioning, thereby delaying zebrafish development [61]. Although it has been demonstrated that embryonic development is influenced by strong SMFs, the left–right asymmetry is closely related to calcium signaling and DNA integrity. Abnormalities in calcium ion levels caused by SMFs can affect the expression of these signals, leading to a range of congenital disorders [87]. Currently established mechanisms include the impact on calcium ion concentration and reactive oxygen species (ROS) levels. One potential study suggests that high levels of ROS may result from the application of Lorentz forces on charged biomolecules [88]. In the study of delayed embryonic development in *Caenorhabditis elegans*, it was found that the specific mechanism by which ultra-high SMFs cause delayed embryonic development is through the induction of spindle abnormalities in early embryonic cells, the downregulation of genes related to asymmetric embryonic division, and the abnormal expression of non-muscle myosin NMY-2 in the division furrows [21].

Some studies have also found no significant harmful effects of static magnetic fields (SMFs) on embryonic development. For instance, Oliva observed that exposing coral larvae to a 1 mT SMF for 48 h did not affect their development [46]. Tian found that after exposing zebrafish embryos to a 22 T SMF for 2 h, the homogenous 22 T SMF did not reduce the hatching rate of zebrafish eggs and had no impact on embryonic development [62]. Therefore, further research is needed to fully understand the impact of SMFs on embryonic development.

## 7. The Impact of SMFs on Offspring Health

The incidence of fetal malformations among live births ranges from 2% to 4%. Congenital anomalies often arise from disruptions or interferences during the developmental process, leading to defects in the fetus. Given the various impacts of SMFs on embryonic development, it is essential to explore their potential teratogenic effects on the fetus [57].

Research by Zaun et al. indicated that exposing the uterus of pregnant mice to 7T or 1.5 T SMF did not result in significant differences in the size of the testes and epididymis in male offspring. Additionally, there were no changes in sperm count, motility, or size [58]. Research shows that, after pregnant rats were exposed to a 0.5–0.7 T SMF, changes were assessed in the testes and epididymis of male offspring. The findings revealed that exposure to SMFs in the uterus did not lead to changes in body weight or the weight of the testes and epididymis [59]. After exposure to a static magnetic field (SMF) of 0.1mT, the number of spermatogonia in rats significantly decreased, along with a marked reduction in the volume of seminiferous tubules and their epithelium [63]. Monfared found that administering vitamin C to rats exposed to a 1.5 T static magnetic field (SMF) for 30 min provided protective effects against sperm damage caused by the SMF [64].

Exposure to a 1.5 T MRI did not result in significant differences in the blastocyst formation rates among in vitro mouse embryos [89]. Zahedi and colleagues found that pregnant mice exposed to the 1.5 T MRI scanning entrance and the 7 T MRI center experienced a slight delay in weight gain and in time to eye opening compared to controls [65]. Cardiac progenitor cells in mouse embryos exhibited an enhanced ability to differentiate into cardiomyocytes when exposed to SMFs [68]. For mice with lipopolysaccharide-induced preterm labor, daily exposure to an SMF for 40 min can delay the onset of preterm labor [66]. Studies suggest that exposure to various SMFs influences the development of otoliths, which are crucial for fish survival, thereby potentially impacting the survival of fish eggs [69,70,71,72]. In their study, the magnetic field was uniform at the center but gradually increased from the entrance to the center. They observed that zebrafish eggs exposed in the center experienced significantly less impact than those at the entrance, with effects increasing linearly [70,71]. In an experiment with fertilized chicken eggs, Nishimura found that exposing the eggs to a 1.1 mT static magnetic field (SMF) resulted in a lower incidence of abnormalities, such as mandibular edema and tailbone defects in the offspring. Moreover, no significant intergroup differences were observed. Therefore, the researcher concluded that exposure to this intensity of SMF does not have any significant teratogenic effects on chick embryos [73]. However, other studies have found that exposing chicken embryos at different developmental stages to a 1.5 T static magnetic field (SMF) increases the rates of developmental abnormalities and mortality. Additionally, exposure to a 0.2 T SMF for three hours suppresses the angiogenic response in chicken embryos [74,90]. Other studies have found that SMF exposure significantly affects the developmental processes of avian embryos, while in mammals, the primary impact has been on skeletal changes [91].

Research on female MRI operators suggests a potential link between MRI exposure and increased rates of preterm birth and low birth weight, though the results did not reach statistical significance [75]. In a study by Choi investigating pregnant women exposed to 1.5T MRI during the first three months of pregnancy, it was found that among 15 live-born infants, 2 were born with congenital malformations [92]. After exposing pregnant mice to SMF, researchers found that various intensities of SMFs may have potential teratogenic effects on offspring. Exposure to 400 mT SMF resulted in significant malformation rates (9.5%) in the offspring. Additionally, embryos exposed to 1.5 T MRI showed a significantly higher rate of eye development malformations compared to the control group. Furthermore, exposing embryos to 10 mT SMF for 20 days increased fetal resorption and mortality, reduced crown–rump length and fresh weight, decreased vascular differentiation, and induced histological changes. This was accompanied by a reduction in the expression of vascular endothelial growth factor (VEGF) protein in several organs, as well as an increase in reactive oxygen species (ROS) levels [67,93,94]. However, a meta-analysis of existing animal studies on prenatal MRI found no evidence to suggest that MRI exposure adversely affects reproductive and offspring outcomes [95].

For embryos still in the uterus, SMFs have potential teratogenic effects, such as causing eye development abnormalities in fetuses. In non-mammals, the rate of malformations and mortality increases with higher SMF exposure. However, experiments have shown that embryos exposed to SMFs in the uterus do not exhibit impaired reproductive organ function after reaching sexual maturity. This may be because reproductive organs continue to develop during sexual maturation. This also suggests that the teratogenic effects of SMFs may be transient [58].

## 8. The Relationship Between SMFs and Infertility

A study suggests that lifelong exposure to a 76.655 mT static magnetic field (SMF) in female mice resulted in no pregnancies, indicating a potential inhibitory effect of strong SMFs on conception [96]. Zaun et al. found that, when pregnant rats were exposed to 1.5 T or 7 T SMF, the number of offspring was not significantly reduced; however, subsequent analysis revealed that the offspring of rats exposed to SMFs had significantly smaller placentas [58]. High et al. reported that pregnant rats exposed to 9.4 T SMF for 60 h showed no adverse effects on pregnancy outcomes, including gestation duration, number of live births, total offspring count, or the sex ratio of the pups [97].

A cross-sectional study involving pregnant women preparing for abortion due to unintended pregnancies found that exposure to an SMF for 24 h resulted in a shortening of the embryo length, thereby inhibiting the embryonic development process. This was accompanied by a higher degree of apoptosis; however, the relationship between SMF exposure and apoptosis was not statistically significant [98]. A cohort study explored the relationship between SMFs and miscarriage, finding that multiple exposures to strong SMFs had a greater impact on women with reduced fertility, increasing the likelihood of miscarriage [99].

## 9. Conclusions

Overall, SMFs have significant effects on various processes of reproductive development (Figure 1). In terms of male reproduction, both 1 mT and 10 T SMF exposure negatively impact sperm activity, leading to reduced motility and potential teratogenic effects. Therefore, it is crucial for males at risk of high SMF exposure to take protective measures, as this could influence occupational health guidelines for professionals working in high-SMF environments.

For female reproduction, ethical considerations have limited most studies to model organisms, such as nematodes, rats, and mosquitoes. Findings indicate that SMF exposure can inhibit the embryonic development of these animals and may also exhibit teratogenic effects, while promoting apoptosis in ovarian and uterine cells. However, some research suggests that SMFs in the range of 20 mT to 60 mT may have a protective effect on oocytes, aiding in the preservation of oocyte viability during freezing. High-intensity SMFs have also been shown to protect mouse ovaries by enhancing antioxidant capacity and improving gut microbiota diversity. Regarding the impact on embryos, various animal studies consistently show that SMF exposure reduces fertility rates, has teratogenic effects, and delays development. However, most of these studies are observational, and research specifically targeting the mechanisms by which magnetic fields induce changes is still lacking. Based on current findings, while some studies suggest that SMF exposure may enhance antioxidant capacity and improve oocyte preservation during cryopreservation, which could contribute to advancements in assisted reproductive technologies, it is still advisable for pregnant women to avoid SMF exposure during gestation to reduce the risk of congenital anomalies in their offspring.

Research has shown that variations in magnetic field strength and exposure duration can lead to significant differences in results. Different locations within the same magnetic field can also affect outcomes; for instance, offspring sperm from pregnant mice placed at the center of an MRI produced normal sperm, while those at the MRI entrance produced abnormal sperm. Similarly, zebrafish larvae at the center of a static magnetic field (SMF) exhibited higher otolith fusion rates than those at the periphery, likely due to differences in field uniformity.

Current studies generally categorize magnetic fields into three types: uneven fields generated by magnets and uniform fields produced by superconducting or conventional magnets. The available evidence suggests that uneven magnetic fields have a more significant impact on reproductive development than uniform fields. Existing literature demonstrates significant differences in experimental outcomes based on varying exposure durations. For instance, in studies with short exposure durations, acute effects, such as changes in cell membrane potential or calcium ion influx, are typically observed. In contrast, longer exposures are more likely to induce chronic effects, such as alterations in gene expression regulation or structural changes in reproductive organs. These differences suggest that the duration of static magnetic field (SMF) exposure plays a critical role in its mechanisms of action. It should be noted that some studies have not reported detailed exposure durations or have not analyzed it as an independent variable, which may limit the interpretability of the results. For instance, certain experiments only document the intensity of exposure and the conclusions, while neglecting the potential influence of the exposure duration on the outcomes or just giving an approximate time instead of an exact time [52,55,60,75,86]. Such omissions can reduce the comparability of results across different studies and increase uncertainties in replicating experiments. To address these limitations, future research should place greater emphasis on recording and analyzing exposure duration as a key variable. Specifically, experiments should include a range of exposure time gradients to investigate whether a “critical effect threshold” exists—a specific time point after which the observed effects significantly intensify or diminish.

The effects of various magnetic field strengths on female reproduction are not yet fully understood, particularly regarding the impact of high-intensity fields, which remains a gap in the research. Studies on embryonic development highlight the potential teratogenic risks of SMF exposure during pregnancy, reinforcing the need for clear guidelines on exposure limits for pregnant women. However, some research suggests that SMF-induced developmental delays may be reversible, underscoring the importance of further studies to establish safe exposure thresholds. While the mechanisms underlying SMFs’ effects on reproductive function remain unclear, evidence indicates that its impact on sperm may be mitigated through antioxidant treatments tailored to individual populations. Overall, the uniformity, strength, duration, and method of exposure to SMFs are crucial factors influencing reproductive toxicity. More data and experiments are needed to clarify the specific impacts of different SMF strengths and exposure durations on reproduction. This research could inform the development of targeted therapeutic approaches based on varying SMF intensities, ultimately guiding the safe application of SMFs in the future.

## Figures and Tables

**Figure 1 cimb-47-00116-f001:**
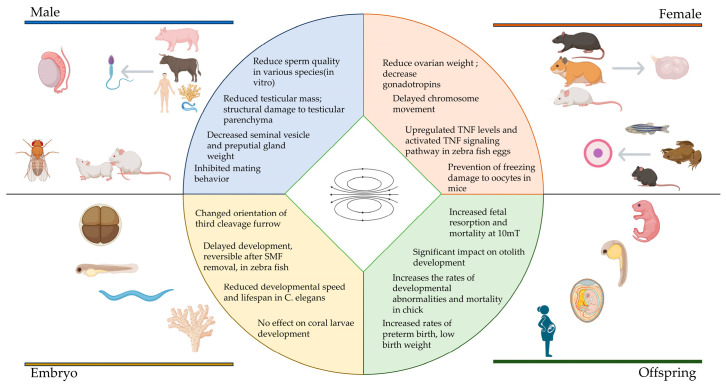
The effect of SMFs on reproduction.

**Table 1 cimb-47-00116-t001:** Effects of SMFs on reproduction.

The Effect of SMF	Species	Magnetic Field Intensity (SMF)	Exposure Time	Effects
On mating behavior	Rats	0.2 mT	22 h/day, 7 days after mating	No statistically significant differences in fertility index and mating frequency [31]
	Mice	4 T	Continuous 7day’s	Inhibited mating behavior [32]
	*Drosophila melanogaster*	5 T	Continuously during the initial five days of adult life	Reduced mating and reproductive behavior by 10.7% [33]
On testes and epididymis tissues	Mice	1.5 T	15 min	Decreased sperm count; slight, statistically insignificant abnormalities in sperm morphology [34]
		1.5 T MRI	36 min/week for 3 weeks	Reduced testicular mass; structural damage to testicular parenchyma; decreased sperm motility [35]
		0.3 T MRI	66 h	No evidence of cytotoxicity [36]
	Rats	25 mT	18 consecutive weeks	Increased LH levels; decreased testosterone levels [37]
		25 μT	90 days	No change in testicular weight but decreased seminal vesicle and preputial gland weight; reduced sperm count [38]
	*C. elegans*	8.5 T	1 h/3 h/5 h	Increased apoptosis in reproductive cells, mitigated by DMSO (antioxidant) [39]
On sperm quality	Human sperm (in vitro)	1 T	3 h	Sperm exhibited magnetotaxis, aligning their long axis perpendicular to the magnetic field [40]
		1 mT	2 h	Decreased sperm motility; increased reactive oxygen species [41]
	Bull sperm (in vitro)	1 T	10 min	Bull sperm aligned perpendicular to the magnetic field [42]
	Pig sperm (in vitro)	1 mT	1 h	Weakened sperm ability to undergo acrosome reaction; negative impact on sperm motility and morphology; disrupted calcium ion homeostasis [43]
	*Caenorhabditis elegans*	10 T	3 h/5 h/10 h	Significant reduction in sperm motility; sperm abnormalities observed [44]
	Mice	0.7 T	35 days (1–2 h/day)	Abnormal hook shape of sperm heads; structural changes [45]
	Coral sperm (in vitro)	1 mT	3 h	Decreased fertilization rates; compromised DNA integrity [46]
On ovarian function	Rats	25 μT	18 weeks	Significant reduction in ovarian weight; decreased gonadotropins (FSH and LH) [47]
	Ovarian cells (hamster)	10 T (non-uniform)	4 days	No effect on cell growth or cycle; increased micronucleus frequency with X-ray radiation [48]
		5 mT	24 h	Delayed chromosome movement [49]
	Mice	150 mT (uniform)	18 weeks (8 to 26 weeks of mice age)	Increased follicle number; enhanced antioxidant capacity; improved microbiome and physical performance [50]
On oocyte quality	Zebrafish eggs	11.4 T	5 days	No significant changes in mortality, hatching rates, or body length; upregulated TNF levels and activated TNF signaling pathway [51]
	Porcine granulosa cells	2 mT	Acute exposure	Decreased mitochondrial activity in follicular cells, impairing oocyte reproductive function [52]
		2 mT	72 h	Weakened mitochondrial activity; reduced progesterone and estrogen production; negative impact on follicular granulosa cells [53]
	African clawed frog eggs	0.5–9.4 T	2 h	Reorganization of cortical pigment [54]
	Mice	20 mT	/	Prevention of freezing damage to oocytes [55]
		60 mT	13 min	Reduced freezing damage; improved functionality and regulation of pluripotency of derived blastocysts [56]
On fertility	Mice	76.655 μT	Lifetime	No pregnancies observed [57]
	Rats	1.5 T, 7 T	Daily in utero	No significant reduction in offspring; smaller placentas in offspring [58]
		9.4 T	60 h	No adverse effects on pregnancy outcomes [59]
On early embryonic development	African clawed frog embryos	8 T	Throughout the experiment	Changed orientation of third cleavage furrow; no effect after cleavage [60]
	Zebrafish embryos	9 T	24 h over 6 days	No survival impact; delayed development, reversible after SMF removal [61]
		22 T	2 h	No reduction in hatching rate or impact on embryonic development [62]
	*Caenorhabditis elegans* embryos	4 T, 10 T, 14 T, 27 T	3 h	Reduced developmental speed and lifespan; spindle abnormalities; gene downregulation [21]
	Coral larvae	1 mT	48 h	No effect on development [46]
On offspring health	Rats	1.5 T, 7 T	Daily in utero	No changes in male offspring reproductive organs; smaller placentas [58]
		0.5–0.7 T	From day 7 of gestation to the day of birth.	No changes in body weight or testes/epididymis weights [59]
		0.1 mT	4 h/day for 52 days	The number of spermatogonia significantly decreased; reduction in the volume of seminiferous tubules and their epithelium [63]
		1.5 T	30 min	Protective effects of vitamin C against sperm damage [64]
	Mice	1.5 T	75 min/day during the entire course of pregnancy	A slight delay in weight gain and in time to eye opening [65]
		2.8–476.7 mT	40 min/day	Fetal development and the delivery were normal in animals that were exposed to SMF but not treated with LPS [66]
		400 mT;1.5 T	6 min/day, from Gd 7.5 to 14.5; Gd 7 for 36 min	Malformation rates of 9.5% at 400 mT; eye abnormalities at 1.5 T [67]
		10 mT	20 days during gestation	Increased fetal resorption and mortality at 10 mT [68]
	Zebrafish larvae	14 T	2 h	Significant impact on otolith development [69,70,71,72]
	Chick	1.1 mT	19 days	Lower incidence of abnormalities in chick; no teratogenic effects at 1.1 mT [73]
		1.5 T	6 h	Increased rates of developmental abnormalities and mortality [74]
	Female MRI operators and pregnant women	1.5 T	/	Increased rates of preterm birth, low birth weight, and congenital malformations [75]

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
