# Peer review of "Recent Studies on the Effects of Static Magnetic Fields (SMF) on Reproductive Function"

_cimb, 2025, doi:10.3390/cimb47020116_

Round 1

Reviewer 1 Report

Comments and Suggestions for Authors

This study focused on reviewing the recent publications regarding the effects of SMF in the field of reproduction, coupled with different SMF intensities and durations on multiple animal models and human clinical studies. This review is well organized and thoroughly investigating the effects of low-intensity, medium to high-intensity, and high-intensity on male and female reproduction, as well as embryo development, respectively, which augments the quality of this study.

The quality of the manuscript is good already only with a few points need to be further edited:

Specific comments:

1. Page4 , table 1, “on oocyte equality” should be “on oocyte quality”.

2. Table1, the position of “On Early Embryonic Development” should be swapped with “On fertility” to make it more logical.

3. Line109, “Further research by the team revealed that SMF decreases reproduction in…”, better to be “Further research by the team revealed that SMF is harmful to reproduction in…”

4. Line225, the sentence “Currently, there are no controlled variable experiments using the same magnetic field generator to vary the intensity significantly.” is confused, please make it clear.

5. Line 230, “on oocyte equality” should be “on oocyte quality”.

6. Line 246, “porcine Gcs cells” should be “porcine GCs”.

7. Line 281-282, you mentioned the development is not affected, but the development rate is affected. Please clarify this in a better way.

8. Based on the structure of this review, section 7 and 8 should be swapped, as early embryo development and offspring health are continuous contents and should not be disturbed by infertility.

9. Line 386-387, by “However, studies indicate that SMF does not affect the reproductive function of embryos in the uterus” are you trying to suggest that SMF does not affect the reproductive organs and gametes development? Please make it clear.

10. Line 391, “Overall, SMF has significant…”.

11. In figure 1, to summarize the whole review, it is better to show the affect of SMF on male, female, embryo, and Offspring, separately. Especially for embryo part, do they performed in vitro or in vivo need to be clarified. For all the species mentioned in each part, it is better to put in this graphic summary as well. Also, as a lot of controversial results exist, you may also show that here.

12. By the end, it is necessary to clarify how those research are meaningful on practice of human clinic, as it is the most important reason people conducting these experiments.

Reviewer 2 Report

Comments and Suggestions for Authors

This review synthesizes findings from animal and clinical studies on the effects of the increasing use of static magnetic fields (SMF) on reproductive function, focusing on varying intensities and exposure durations, emphasising mitochondrial function, chromosomal division, and embryonic development. Despite growing research, the influence of SMF on reproductive function remains debated, necessitating further investigation. Research gaps remain, particularly concerning the mechanisms of SMF-induced changes. This review establishes a foundation for further research into SMF's effects on reproductive function, calling for more comprehensive studies to assess safety and applications.

Overall, the topic is interesting, the information provided is detailed and well-organized in paragraphs and subparagraphs and the references cited are pertinent. However, I suggest some slight changes to improve the manuscript.

- Throughout the text, the name of the leading researcher is repeated for each discovery. Sometimes it is also useful to do so, but not so often. Always putting names makes the text less fluent. I will give an example to better explain the concept: in line 68, instead of writing "in the previous review Song C..." you can write "Previous studies have shown...", and then just put the specific reference at the end of the sentence. I believe that by making these changes throughout the text, the manuscript would become easier to read.

- Table 1 is very useful and summarizes well the effects of SMF on the reproduction of various animals, however since there are many columns, to make it less crowded, I would suggest removing the "Researchers" column and moving the references (indicated by numbers) next to each of the specific effects.

- Lines 178-180 are a repetition of what has already been extensively stated before. The paragraph is not so long as to require a summary, so it can be removed.

- The title of subparagraph 5.2 and also in Table 1 should be "oocyte Quality".
